# Non-hysteretic first-order phase transition with large latent heat and giant low-field magnetocaloric effect

F. Guillou[1], A.K. Pathak[1], D. Paudyal[1], Y. Mudryk [1], F. Wilhelm[2], A. Rogalev[2] & V.K. Pecharsky[1,3]

First-order magnetic transitions (FOMTs) with a large discontinuity in magnetization are highly sought in the development of advanced functional magnetic materials. Isosymmetric magnetoelastic FOMTs that do not perturb crystal symmetry are especially rare, and only a handful of material families, almost exclusively transition metal-based, are known to exhibit them. Yet, here we report a surprising isosymmetric FOMT in a rare-earth intermetallic, $Eu_2In$. What makes this transition in $Eu_2In$ even more remarkable is that it is associated with a large latent heat and an exceptionally high magnetocaloric effect in low magnetic fields, but with tiny lattice discontinuities and negligible hysteresis. An active role of the Eu-5$d$ and In-4$p$ states and a rather unique electronic structure borne by In to Eu charge transfer, altogether result in an unusual exchange mechanism that both sets the transition in motion and unveils an approach toward developing specific magnetic functionalities ad libitum.

[1] The Ames Laboratory, U.S. Department of Energy, Iowa State University, Ames, IA 50011-2416, USA. [2] ESRF, The European Synchrotron, 71 Av. des Martyrs, 38000 Grenoble, France. [3] Department of Materials Science and Engineering, Iowa State University, Ames, IA 50011-1096, USA. Correspondence and requests for materials should be addressed to F.G. (email: francoisguillou@imnu.edu.cn)

In the solid state, phase transitions commonly lead to emergence of important properties such as superconductivity or ferroelectricity, or to a change in magnetic or crystallographic order[1]. In fact, phase transitions are at the heart of many functionalities. Among them, magnetic phase transitions enable a plethora of both established (magnetostrictive, magnetic shape memory, magnetoresistance) and emerging (magnetocaloric) applications[2–6]. The latter rely on the magnetocaloric effect (MCE), currently an intensely researched subject for its use in heat pumping systems, which promises transformative improvements in heating/cooling technologies, making them both energy-efficient and environmentally-benign[2–6]. In addition to obviously important applications in the room temperature range, magnetic refrigeration is also expected to create new opportunities in the low or ultra-low temperature ranges, including liquefaction of natural gas and hydrogen, and in space.

Consider a hierarchy of phase transitions illustrated in Fig. 1, focusing on those that involve magnetism. Based on the Ehrenfest classification, a distinction is usually made between the most common continuous second-order transitions, and the far less frequent discontinuous first-order magnetic transitions (FOMTs). The latter are defined in practice by the presence of a latent heat of transformation. Further, discontinuities in the entropy ($\Delta S$), volume ($\Delta V$), magnetization ($\Delta M$), and other measurable quantities can be induced by the conjugate driving field(s), forming the basis for functionality of materials exhibiting FOMTs. Among those, two classes of FOMTs can be distinguished. The most frequent are magnetostructural transitions involving a simultaneous change in crystal symmetry, whereas the magnetoelastic FOMTs occur without changing the symmetry. Magnetoelastic FOMTs may be further classified as those with purely isotropic (e.g., FeRh, La(Fe,Si)$_{13}$) or anisotropic (e.g., (Mn,Fe)$_2$(P,As,Si)) discontinuities in their lattice parameters[7–12]. For magnetoelastic FOMTs common side effects related to hysteresis and irreversibility can be controlled and tuned by compositional adjustments and, compared to magnetostructural transitions, such adjustments are typically more successful in the design of real materials.

Very few material systems are known to undergo magnetoelastic FOMTs at ambient conditions with at least one phase having a large magnetization. This lack of representatives is primarily attributed to the unique underlying mechanism of magnetoelastic FOMTs. Most often these transitions involve simultaneous and cooperative discontinuities in the electronic, lattice, and magnetic properties. The most characteristic example is the FOMT between antiferromagnetic (AFM) and ferromagnetic (FM) states in FeRh[8]. The isostructural character of this transformation was established more than half a century ago[7], yet FeRh keeps attracting a great deal of applied and fundamental interest. Though widely debated, the origin of the AFM↔FM FOMT in FeRh is most often ascribed to an evolution of the electronic density of states (DOS) near the Fermi level and an instability of Rh magnetic moments across the transition. La(Fe,Si)$_{13}$ and Fe$_2$P materials are other archetypal examples of magnetoelastic FOMTs[9,10,13,14]. Regardless of certain aspects that may be unique to every known family of materials, magnetoelastic FOMTs in all of them are rooted in drastic electronic reconstructions made possible by the itinerant $d$-states of transition metals, in turn enhancing the magnetic discontinuities. In rare-earth based materials, on the other hand, the established paradigm is that the 4$f$ electronic states of lanthanides, while critical for the development of the largest magnetic moments and strongest magnetocrystalline anisotropies that exist in nature, are highly localized and ineffective in promoting magnetoelastic FOMTs without assistance from itinerant electrons. The latter are usually provided by transition metals whose magnetic exchange interactions spread much wider compared to lanthanides.

Here, we report on the observation of a first-order magnetoelastic transition with an entirely different mechanism in a rare-earth intermetallic, Eu$_2$In. After establishing the first-order character of the paramagnetic (PM) ↔ FM transition by measurements of physical properties, including magnetization, heat

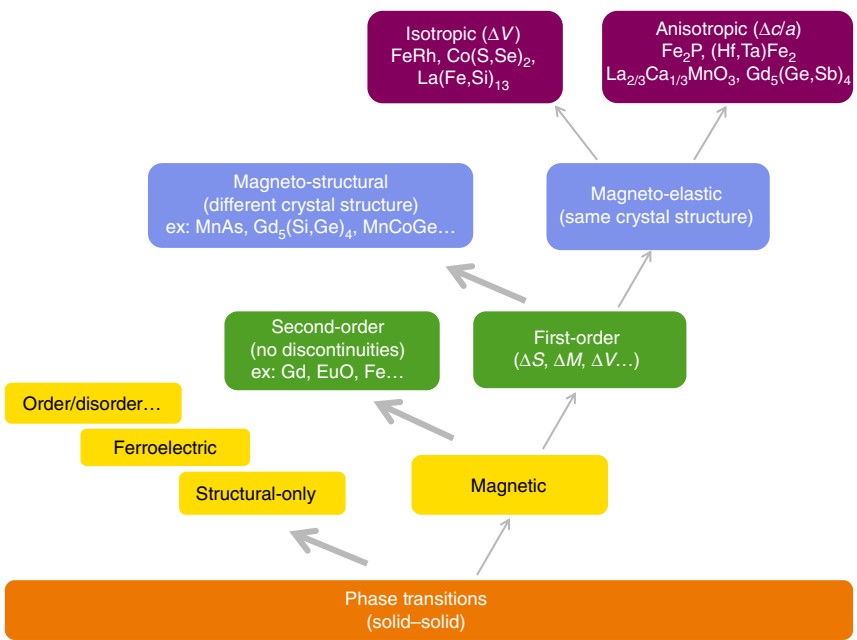

**Fig. 1** A classification diagram of solid-state phase transitions. Several types of magnetic phase transitions are categorized focusing on the magnetoelasticity branch, with examples of materials families[7,9–14,24,38,52–55]. Different sub-classes are distinguished depending on the Ehrenfest classification (first/second order), whether the magnetic transition is associated with a change in crystal symmetry (magneto-structural/magneto-elastic), and depending on the nature of the lattice parameter discontinuities (isotropic/anisotropic)

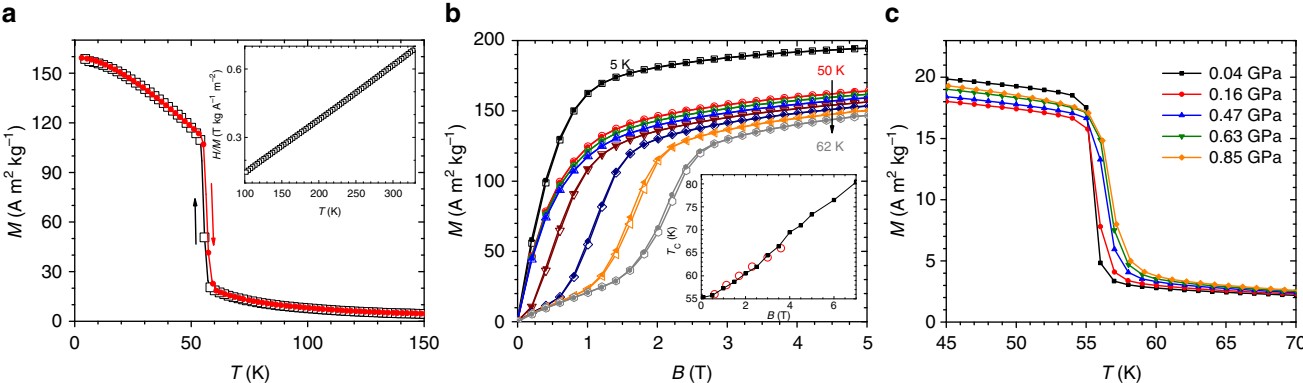

**Fig. 2** Magnetic properties of Eu$_2$In. **a** Magnetization ($M$) as a function of the temperature ($T$) measured upon cooling and heating in a magnetic field ($B$) of 1 T. The inverse magnetic susceptibility in $B = 0.1$ T in the paramagnetic regime is displayed in the inset. **b** Isothermal magnetization data measured upon magnetization (open symbols) and demagnetization (closed symbols) for different temperatures: 5 K (squares), then from 50 to 62 K in 2 K increments. The inset displays evolution of the Curie temperature ($T_C$) as a function of the magnetic field derived from isothermal $M(B)$ (circles) and isofield $M(T)$ measurements (squares, raw data not shown). **c** Temperature dependence of the magnetization measured at different applied pressures upon heating in $B = 0.1$ T

capacity, and electrical resistivity, we address the microscopic mechanisms at its origin by temperature dependent x-ray diffraction, emission and absorption, magnetic circular dichroism, as well as first-principles calculations.

## Results

**Eu$_2$In, an exception within the R$_2$In material family**. All R$_2$In binary materials, where R = rare earth element, crystallize in the hexagonal Ni$_2$In type with space group $P6_3/mmc$, except when R = Eu or Yb (ref.[15]). The magnetic properties of hexagonal R$_2$In materials are well documented, in particular for Gd$_2$In, which exhibits a sequence of AFM and FM transitions upon heating[16]. On the other hand, the crystal structure of Eu$_2$In is known but no physical properties were reported so far. A polycrystalline Eu$_2$In prepared by induction melting and annealing was reported to crystallize in the Co$_2$Si-type orthorhombic structure with space group $Pnma$ (ref.[17]). Our sample, synthesized by melting and annealing in a sealed Ta crucible filled with inert gas and using a conventional resistance furnace, adopts the same Co$_2$Si-type crystal structure with lattice parameters at room-temperature $a = 7.4536(6)$ Å, $b = 5.5822(5)$ Å and $c = 10.312(1)$ Å that are close to reported earlier[17], and $a/c$ and $b/a$ ratios typical of other representatives of this type of structure[18]. Additional details about the crystal structure and its characterization can be found in Supplementary Notes 1 and 2.

**Discontinuous FM transition**. The magnetization of Eu$_2$In is presented in Fig. 2. Upon cooling, magnetization reveals a particularly sharp (for a 1 T magnetic field) discontinuity at the Curie temperature $T_C(B = 1 \text{ T}) = 58$ K corresponding to the development of a FM phase via FOMT (AC magnetic susceptibility in Supplementary Note 3 indicates zero DC magnetic field $T_C$ of 55 K). The thermal hysteresis is however very small (~0.07 K from magnetization measurements, Supplementary Note 4). In the range $100 < T < 300$ K, the magnetic susceptibility follows Curie–Weiss law with Weiss temperature of 48 K and effective magnetic moment of 8.5 $\mu_B$ Eu$^{-1}$. Though larger than the theoretical value for non-interacting Eu$^{2+}$ (7.94 $\mu_B$), this indicates that both of the independent Eu-I and Eu-II sites in this intermetallic compound are populated by divalent europium. The high magnetic field magnetization measurements at 5 K shown in Fig. 2b produce a saturation magnetization of at least 14.4 $\mu_B$ f.u.$^{-1}$, in line with a fully collinear ferromagnetism, the divalent character

of Eu ($gJ = 7$ $\mu_B$, where $g$ is gyromagnetic ratio and $J = S = $ total angular momentum quantum number of Eu$^{2+}$), and a small but measurable additional magnetic contribution from the 5$d$ states of Eu, even though theoretically, Eu$^{2+}$ is expected to have a [Xe] $4f^76s^25d^0$ configuration. Above the Curie temperature, the metamagnetic jumps observed in the isothermal measurements indicate nearly fully reversible magnetic field-induced PM↔FM transitions with a nearly negligible (≤0.1 T) magnetic hysteresis, in line with the extremely narrow thermal hysteresis.

The magnetic field dependence of $T_C$, determined from both the isothermal and isofield measurements, is presented in the inset of Fig. 2b. Both sets of data are in good agreement, leading to $\partial T_C/\partial B \approx +3.5$ K T$^{-1}$, which is in line with $T_C = 55$ K determined from AC measurements in 0.0005 T AC field (Supplementary Note 3) and $T_C = 58$ K determined from the 1 T data presented in Fig. 2a. Using $\Delta M \cong -90$ A m$^2$ kg$^{-1}$ determined from isothermal magnetization data at $T = 58$ K and Clausius–Clapeyron formalism $\Delta S_T = -\Delta M(\partial B/\partial T_C) = \Delta V (\partial P/\partial T_C)$, the estimated entropy change at the transition is $\Delta S_T \cong 26$ J kg$^{-1}$ K$^{-1}$ and the latent heat $\Delta L = T\Delta S_T \cong 1.4$ kJ kg$^{-1}$. The application of hydrostatic pressure has a minor effect on $T_C$, and the unmodified saturation magnetization indicates that the divalent character of Eu is preserved up to 0.85 GPa (Supplementary Note 5). The shift of the ordering temperature $\partial T_C/\partial P$ of about $+2$ K GPa$^{-1}$ points to a small and positive cell volume change at the FOMT upon heating that too can be estimated from the Clausius–Clapeyron equation as $\Delta V/V \approx 0.03\%$. Common for many other pressure-sensitive FOMTs, the transition width increases with the pressure by ~20%, as derived from the full width at half maximum of $\partial M/\partial T$ curves. As expected, the FOMT in Eu$_2$In also leads to a sharp drop in electrical resistivity upon cooling with a narrow thermal hysteresis of about 0.1 K separating two regions with metallic behaviors (Supplementary Note 6).

**Large latent heat without hysteresis**. One of the most important quantities characterizing a first-order transformation—which actually is at the basis of the Ehrenfest classification—is its latent heat. However, for extremely sharp and hysteretic transitions heat capacity measurements are always challenging[19]. Hence, we combined the standard heat capacity data provided by a commercial semi-adiabatic calorimeter with an external analysis of a single relaxation across the FOMT (referred to as single pulse

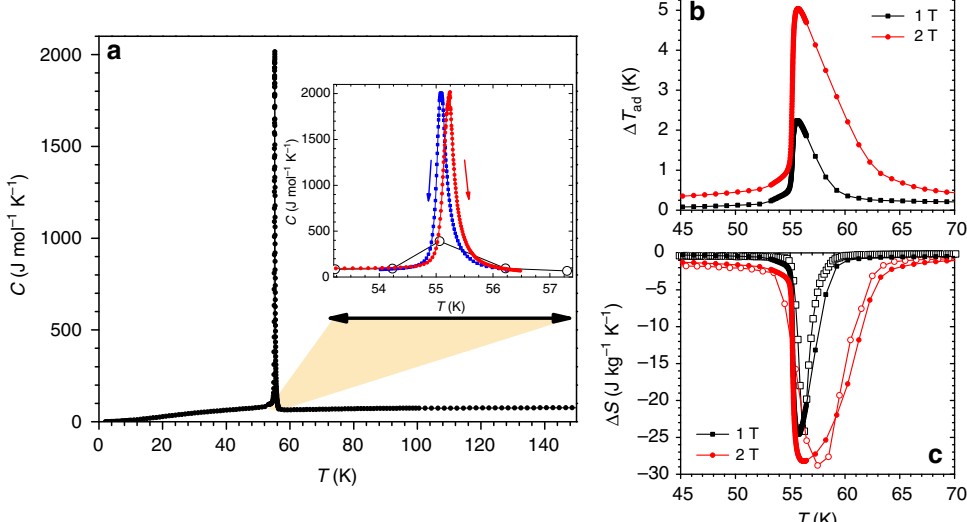

**Fig. 3** Heat capacity and magnetocaloric effects of Eu$_2$In. **a** Heat capacity ($C$) as a function of the temperature ($T$) measured on heating in zero magnetic field. The inset shows the heating and cooling branches calculated using the single pulse method (SPM, filled squares) compared to the standard analysis of the Multiview software Quantum Design (open circles). **b** Adiabatic temperature change ($\Delta T_{ad}$) for magnetic field changes of 1 and 2 T derived from calorimetry. **c** Isothermal entropy change ($\Delta S$) determined from calorimetry (filled symbols) and magnetization (open symbols)

method, SPM, Supplementary Note 7)[20]. Figure 3 shows the extremely sharp heat capacity peak of Eu$_2$In associated with the FOMT (the full width at half maximum is less than 1 K), and the data in the inset confirm the very small thermal hysteresis of about 0.1 K. The extremely high heat capacity maximum and the symmetrical shape of the peak unambiguously establish the first-order character of the transition. To the best of our knowledge, a nearly ideal FOMT in Eu$_2$In is only a second such case observed among metallic solids, the first is the FM–AFM transformation in a very high-purity dysprosium metal[21]. This behavior can be ascribed to excellent compositional homogeneity indicating that Eu$_2$In is a true line compound without a detectable homogeneity region, high crystallinity and low concentration of defects[22,23], and/or low energy barriers involved due to limited interfacial stresses between the two phases. Using a step function to estimate the background in the transition region, the entropy change of the transition $\Delta S_T = \int_{T=54}^{T=56} \frac{(C-C_{background})}{T} dT \cong 27.5$ J kg$^{-1}$ K$^{-1}$ is in line with the estimate made from magnetization measurements using the Clausius–Clapeyron equation. As expected for a ferromagnet[24], with the increase of the magnetic field the heat capacity peak shifts to higher temperature and broadens (Supplementary Note 8).

**Giant MCE at low magnetic field change.** In-field heat capacity measurements were performed to determine the two main quantities of the MCE: the isothermal entropy change $\Delta S$ and the adiabatic temperature change $\Delta T_{ad}$. The maximum adiabatic temperature change is 2.2 and 5.0 K for $\Delta B = 1$ and 2 T, respectively, starting from $B = 0$. These values place Eu$_2$In on top among the best materials available for low-temperature magnetocalorics at present[3–6], especially considering that in this temperature range the lattice entropy of the material is rising very rapidly, naturally suppressing $\Delta T_{ad}$ (ref.[25]). Due to the nearly discontinuous transition and its large sensitivity to the magnetic field, most of the latent heat is converted into a MCE even by a limited magnetic field change of 1 T. The maximum $\Delta S$ values derived from calorimetry are $-24.4$ and $-28.2$ J kg$^{-1}$ K$^{-1}$, for $\Delta B$ of 1 and 2 T, respectively, and are extraordinarily high as well. A good agreement is observed between the $\Delta S$ results derived from

isofield magnetization and calorimetry data. In magnetic fields exceeding 2 T (Supplementary Note 8) the maximum $\Delta S$ begins to saturate, reaching $-37$ J kg$^{-1}$ K$^{-1}$ in 7 T. The steady but slow increase of $\Delta S$ with field above 2 T is due to the conventional spin contribution, but the discontinuity of $\Delta S$ observed for all $\Delta B \geq 2$ T remaining constant around $\sim -26$ J kg$^{-1}$ K$^{-1}$ which, as expected[26], is very close to the entropy change associated with FOMT determined from both the Clausius–Clapeyron equation and from heat capacity. For a field change of 1 T, the coefficient of refrigerant performance (CRP) is about 0.70 and the temperature averaged entropy change (TEC3) is about $-17.1$ J kg$^{-1}$ K$^{-1}$ [27,28]. These figures of merit are comparable to or higher than the most promising magnetocaloric materials in the cryogenic regime[27,28]. With the exceptionally small hysteresis, particularly large $\Delta S$ and $\Delta T_{ad}$ in moderate magnetic fields, Eu$_2$In outperforms all known magnetocaloric materials in this temperature range[3–6].

**Unusual mechanism of FOMT in Eu$_2$In.** The unit cell parameters and crystal structure of Eu$_2$In are presented in Fig. 4. The orthorhombic primitive unit cell consists of four formula units with two inequivalent Eu sites, both located in 4c positions. The Eu-I site (0.0287, 1/4, 0.7048) is surrounded by 6 Eu-II, 2 Eu-I, and 4 indiums atoms. The Eu-II site (0.1781, 1/4, 0.0680) is in a similar environment with slightly different interatomic distances. The Eu$_2$In structure is effectively built by two alternating flat atomic layers stacked along the $b$-axis, where one layer is a mirror image of another shifted diagonally in the $ac$ plane (Supplementary Notes 1 and 2). However, there is no significant difference between the interlayer and intralayer atomic distances suggesting that this is clearly a 3D structure.

To get experimental insight into the microscopic mechanisms of the FOMT in Eu$_2$In and in particular to examine potential structural changes, powder x-ray diffraction experiments were carried out as a function of temperature. A cursory visual examination of the collected powder XRD patterns reveals only minor changes upon cooling (Supplementary Note 2) and the same crystal structure over the whole examined temperature range, indicating an isostructural transition. According to Rietveld refinements, the FOMT in Eu$_2$In corresponds to a small

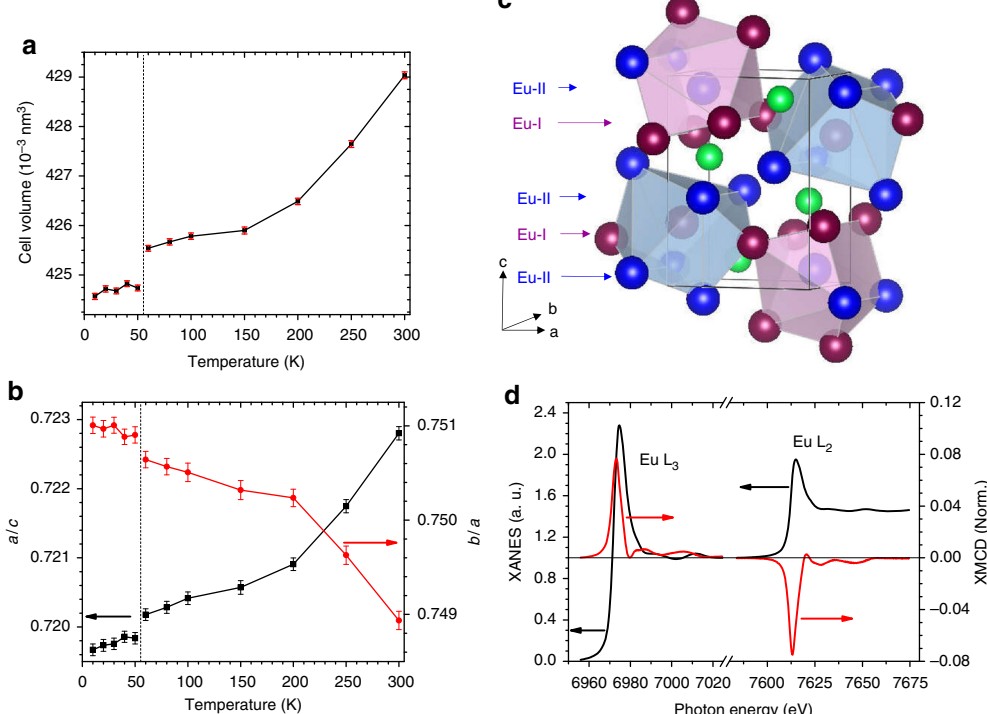

**Fig. 4** Crystal structure and Eu electronic configuration in Eu$_2$In. **a** Unit cell volume ($V$), **b** ratio $c/a$ and $b/a$ of the lattice parameters, determined from powder x-ray diffraction as function of the temperature. The error bars represent the standard deviation estimated during the refinement of the lattice parameters. **c** Schematic representation of the crystal structure. **d** X-ray absorption near edge spectroscopy (XANES) and normalized magnetic circular dichroism (XMCD) spectra at Eu L$_{2,3}$ edges at $T = 5$ K and $B = 3$ T

discontinuous expansion of all lattice parameters: $\Delta a/a \cong 0.08\%$, $\Delta b/b \cong 0.04\%$ and $\Delta c/c \cong 0.03\%$. The resulting unit cell volume change is small and positive, $\Delta V/V \cong +0.1\%$, which is slightly larger than the rough estimate made from high pressure magnetization experiments. In comparison to the most representative examples of magnetoelastic FOMTs (FeRh, La(Fe,Si)$_{13}$, MnFe(P,Si)), the transition in Eu$_2$In manifests an order of magnitude smaller discontinuities in lattice parameters and phase volume[7,9,12]. This is quite surprising, as it is usually believed that either or both significant volume changes or large anisotropic distortions are required to get both a strong FOMT and a large total entropy discontinuity $\Delta S_T = \Delta S_M + \Delta S_{st}$ (where $\Delta S_M$ and $\Delta S_{st}$ reflect perturbations in the magnetic and crystallographic sublattices, respectively[29]) simply because $\Delta S_{st}$ scales proportionally to $\Delta V/V$ (ref.[30]). On the other hand, the similarity between the lattices of FM and PM phases is the key ingredient underpinning the extremely small thermal hysteresis of the transition, requiring a minimum free energy difference to overcome strain energy. The transition in Eu$_2$In is also in stark contrast with other types of solid–solid phase transitions such as martensitic transformations, which usually requires a fine tuning of the cell parameters to ensure lattice compatibility and approach reversibility[31,32].

The actual $\Delta V/V \cong 0.1\%$ from the x-ray data (0.03% from the Clausius–Clapeyron equation) is too small to result in both the extraordinary large latent heat and the giant small-field MCE in Eu$_2$In. Hence we must consider the magnetic entropy change across $T_C$. The total magnetic entropy available in Eu$_2$In is $S_M = R\ln(2J + 1) = 17.3$ J mol(Eu$^{2+}$)$^{-1}$ K$^{-1}$ = 82.6 J kg(Eu$_2$In)$^{-1}$ K$^{-1}$, where $R$ is the universal gas constant. When compared to the actually observed discontinuity $\Delta S_T \cong 26$–27.5 J kg$^{-1}$ K$^{-1}$, the compound demonstrates an exceptionally high, nearly 30%

concentration of the total available spin entropy at $T_C$. Most importantly, this large amount of entropy can also be reversibly shifted above $T_C$ by a magnetic field change as small as 1 T, generating a giant small-field MCE. This value is indeed remarkable since for comparison, a 1 T magnetic field induces an order of magnitude smaller $\Delta S$ in elemental Gd that corresponds to only ~3% of the available $S_M$, and magnetic field change of approximately ~30 T is required to reach $\Delta S$ on the order of 30% of the available entropy[33,34].

The small $\Delta V$ also makes it highly unlikely that the FOMT is associated with a Eu valence change. To confirm this, x-ray absorption near edge spectroscopy (XANES) and magnetic circular dichroism (XMCD) experiments, carried out at the europium L$_{2,3}$ edges ($2p \rightarrow 5d$) as a function of the temperature, are shown in Fig. 4d. The XANES spectra at $T = 5$ K present a single white line shape with a maximum at ~6974 eV at the Eu L$_3$ edge. Both L$_3$ and L$_2$ spectra are typical of divalent-like Eu, which brings support to the analysis of the magnetization data. Determining the valence by the conventional spectral decomposition into Lorentzian peaks and arctangent (assuming that the broad feature about 7 eV above the white line can be attributed to Eu$^{3+}$)[35], an average valence of +2.03 for Eu was obtained. XANES spectra were measured as function of temperature, and confirm the absence of valence fluctuations and drastic changes in the local crystalline environment of the europium atoms (Supplementary Note 9). The sizable XMCD signals observed at both Eu L$_2$ and L$_3$ edges are comparable with more ionic europium materials whose divalent Eu$^{2+}$ configuration is well established[36], unambiguously demonstrating the FM order in Eu$_2$In and that Eu-4$f$ states carry about 7.0 μ$_B$ Eu$^{-1}$. The saturation magnetization larger than 14 μ$_B$ (Eu$_2$In)$^{-1}$ should accordingly be ascribed to non-negligible

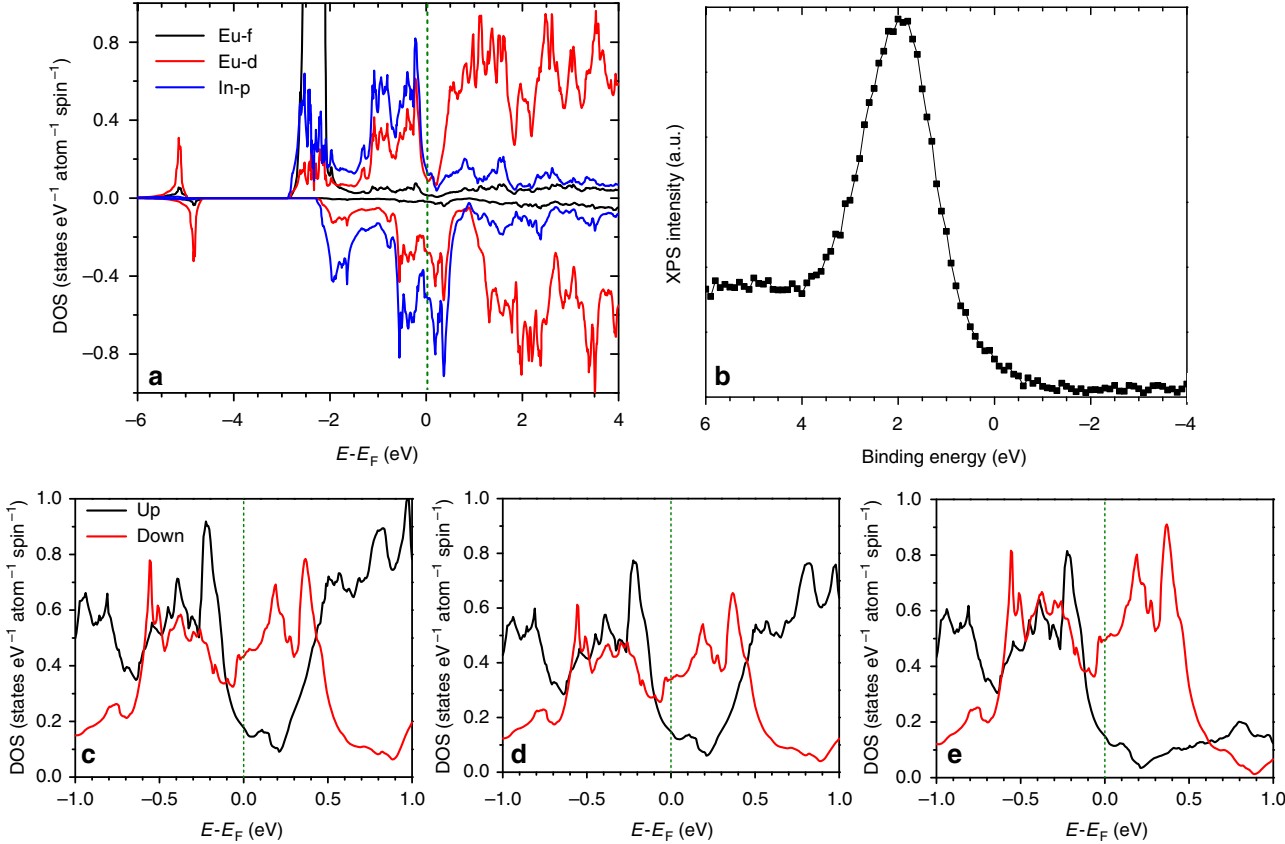

**Fig. 5** Electronic structure of $Eu_2In$. **a** Partial density of states (DOS) of Eu-I and In. **b** Valence band x-ray photon emission spectroscopy (XPS) measurements on $Eu_2In$. Total DOS near $E_F$ (**c**) for Eu-I, **d** Eu-II, and **e** In atoms, in this energy range the DOS are dominated by Eu-5$d$ and In-5$p$ states

magnetic moments on Eu-5$d$,6 $s$ and In-5$p$ states. The XMCD of $Eu_2In$ appears similar or larger than other europium intermetallics such as $Eu_8Ga_{16}Ge_{30}$ (ref.[37]), but due to the inapplicability of optical sum-rules to derive the 5$d$ magnetic moment for rare earths, a more quantitative estimate of the magnetic polarization of the conduction states will be gained through electronic structure calculations.

**Specific electronic structure of $Eu_2In$.** FOMTs in solids with no change of symmetry and no substantial rearrangement of atomic bonding are rare. They have been observed in several transition-metal based compounds, but the underlying mechanisms are often challenging to address. Examples of magnetoelastic FOMTs based on rare-earth are even scarcer. One example we are aware of is $Gd_5Ge_{3.5}Sb_{0.5}$, which shows a significant volume discontinuity without change in crystal symmetry and only small changes in interatomic distances between two polymorphs[38]. Though quite unique (no bond breaking or atomic layer displacements), the mechanisms at the origin of the FOMT remain specific to this 5:4 class of materials[24,39]. To get insight into the origin of the magnetism and the unusual FOMT in the virtually unexplored $Eu_2In$, we turned toward electronic structure calculations.

FM calculations are shown in Fig. 5. Alternative calculations with opposite magnetic moments of Eu-I and Eu-II (an almost zero net magnetization used to approximate the PM state) are provided in Supplementary Note 10. Total energy of FM-$Eu_2In$ is lower by 137.7 meV f.u.$^{-1}$ than the approximated PM-$Eu_2In$, confirming the FM ground state. The 6 $s$, 6$p$, and 5$d$ states of Eu are nearly completely hybridized with the more populated 5$p$

states of In, in such a way that the Eu-$s,p,d$ states fall within the envelope of the In-$p$ states. Due to the strong hybridization of the electronic states in $Eu_2In$, the Eu-5$d$ states become partially occupied (~0.8 e Eu$^{-1}$), rather than being empty as would have been anticipated for Eu$^{2+}$ in the $4f^75d^0$ configuration. The Eu-4$f$ states are centered just below −2 eV, and also hybridize with the $p$ states of In and, consequently, Eu-$s,p,d$ states. Thus, the Eu-4$f$ states in $Eu_2In$ are located deeper than in most EuTMX ternaries with TM a transition metal and X a $p$-block element. This is in line with the absence of mixed valence or valence transition of Eu in $Eu_2In$ but in contrast to EuTMX materials[40,41].

The electronic structure of $Eu_2In$ also contrasts with the closely related $Gd_2In$, where the Gd 4$f$ states are located much deeper[42,43]. X-ray photoelectron spectroscopy measurements (at room temperature), Fig. 5b, show a peak at binding energy of about 2 eV, which is typical of rare-earth 4$f$ states[44] and bring direct experimental support to our predicted energy position for Eu-4$f$ in $Eu_2In$. The Fermi level ($E_F$) is located on the edge of a pseudo-gap in the majority channel, while it corresponds to high Eu-5$d$ and In-5$p$ DOS in the minority channel. In the approximated PM state, $E_F$ falls inside a minor valley surrounded by high Eu-5$d$ and In-5$p$ DOS. High DOS near $E_F$ is often believed to favor instabilities in the electronic structure and magnetism when external parameters, such as temperature, magnetic field, or pressure are altered. In particular, electronic configurations which may lead to significant DOS changes at $E_F$ (and its vicinity) are considered to be at the origin of the magnetoelastic FOMTs in most prototypical transition metal-based compounds[10,14,45].

| Table 1 Magnetic moments on the different orbitals of ferromagnetic $Eu_2In$ | | | | |
|---|---|---|---|---|
| $Eu_2In$ (FM) | s | p | d | f |
| Eu-I | 0.04 | 0.10 | 0.24 | 6.98 |
| Eu-II | 0.03 | 0.06 | 0.20 | 6.96 |
| In | −0.03 | 0.25 | 0.02 | |

The magnetic moments are given in $\mu_B$ atom$^{-1}$

Table 1 shows the magnetic moments in FM calculations. In contrast to the approximated PM calculations, there is a development of a strong exchange splitting on the hybridized Eu-5$d$ and In-5$p$ states in the FM state. Nearly identical (~0.67 eV) Eu-5$d$ and In-5$p$ exchange splitting across the Fermi level leads to similar magnetic moments of 0.22 and 0.24 $\mu_B$ atom$^{-1}$, on Eu-5$d$ and In-5$p$ states, respectively. These values are in line with the experimental saturation magnetization, when the Eu-4$f$ moment (~7.0 $\mu_B$ atom$^{-1}$) is added, and with the large XMCD signal observed at the Eu $L_{2,3}$ edges (2$p \rightarrow$ 5$d$). As argued below, this unusually large Eu-5$d$ and In-5$p$ exchange splitting is likely the key ingredient for the emergence of first-order PM↔FM transition in $Eu_2In$.

Stoner criterion, calculated as the product of exchange splitting (in eV) at the Fermi level and DOS (in states eV$^{-1}$ atom$^{-1}$), is 0.76 for Eu-I 5$d$ states—close to the unity threshold—indicating that the Eu-5$d$ moments are metastable and may act as a trigger for the transition. The important role, the peculiar electronic structure of $Eu_2In$ plays, can be further exemplified by a stark contrast between FM $Eu_2In$ and AFM $Gd_2In$ and $Eu_2Sn$ compounds with similar compositions and related crystal structures. In the hexagonal $Gd_2In$, the Gd-5$d$ states develop magnetic polarization, but the In-$s$ and In-$p$ states are not polarized, and therefore play little, if any, part in its magnetism[42,43]. In AFM $Eu_2Sn$ that adopts the same crystal symmetry as $Eu_2In$, the addition of a single $p$ electron on Sn shifts the whole DOS so that $E_F$ falls into a deep pseudo-gap, Eu-5$d$ and Sn-5$p$ magnetic polarization weakens considerably, hence there is no first-order transition or ferromagnetism (see Supplementary Note 11 for more details on $Eu_2Sn$).

In summary, an intriguing mechanism resulting in a magnetoelastic first-order FM transition is discovered in $Eu_2In$. This transition is associated with a large latent heat while thermal and magnetic hystereses, as well as lattice discontinuities are extremely small, which leads to a remarkable combination of properties. With $Eu_2In$ we show that instabilities leading to FOMTs can be realized in rare-earth intermetallics in the absence of any structural or valence fluctuations. Electronic structure calculations indicate that key elements for the $Eu_2In$ FM transition are the filling of the Eu-5$d$ states by hybridization with other Eu states and In-5$p$ states, and the development of a large exchange splitting on Eu-5$d$ and In-5$p$ states. The strong hybridization between Eu-5$d$ and In-5$p$ states near the Fermi energy affects the long-range exchange interaction between Eu atoms, triggering the FM transition. Demonstrating a strong interplay between a $p$-block metal or a metalloid and rare-earth elements not only broadens a very exclusive class of materials, but it also highlights a favorable route to generate magnetoelastic FOMTs and paves the way for further developments in functional magnetic materials.

## Methods

**Sample synthesis**. Polycrystalline $Eu_2In$ samples were prepared by melting stoichiometric quantities of elemental starting materials followed by annealing in a conventional resistive furnace. The high-purity europium metal was supplied by the Materials Preparation Center of the Ames Laboratory. All operations and handling of the samples were carried out in Ar glovebox. The starting materials were sealed in a Ta crucible under a partial atmosphere of ultrapure Ar. The crucible was then sealed in a quartz tube backfilled with helium gas. The sample was first melted several times at 900 °C in a resistance furnace, flipping sample tube between each melting. Then the sample was annealed at 650 °C for 24 h. The resulting sample has a grey metallic appearance, and is relatively brittle. Exposure to air leads to darkening of the surface within a few hours. The results presented in this communication originate from the same sample, but different samples prepared to ensure reproducibility demonstrate comparable properties.

**Crystal structure characterization**. Temperature-dependent powder x-ray diffraction experiments were carried out on a rotating anode Rigaku TTRAX system using Mo $K_\alpha$ radiation. This system was modified to reach low temperatures using a cold finger He cryostat and to allow in situ application of magnetic field[46]. The powder consisting of particles smaller than 22 $\mu$m was mixed with petroleum jelly, then deposited on a copper sample holder. Room temperature XRD characterization was performed using a PANalytical diffractometer employing Cu $K_{\alpha 1}$ radiation. The powder XRD patterns were refined by the Rietveld method using the Fullprof software. The VESTA software has been used for structure visualization[47]. X-ray photon electron spectroscopy (XPS) measurements were performed at room temperature after in situ cleaning by ion bombardment of the surface of a bulk piece of $Eu_2In$ in ultra-high vacuum.

**Physical property measurements**. The magnetic measurements were carried out in a 7 T magnetic property measurement system MPMS (Quantum Design) magnetometer equipped with a reciprocating sample option (RSO). A quartz straw was used as a sample holder (preparation in Ar glovebox). The MCE up to 7 T was derived by applying the Maxwell equation to isofield magnetization data. The magnetic measurements under pressure were carried out using a Cu-Be mechanical cell manufactured by HMD (type CC-SPr-8.5D-MC4). The inner diameter of the cell was 2.2 mm, and lead was used as an internal manometer. Electrical resistivity measurements were performed using the AC transport option of a 14 T physical property measurement system PPMS (Quantum Design). The heat capacity measurements were performed using the heat capacity option of the same 14 T PPMS system. The sample was mounted using Apiezon N grease. Outside the transition region, the measurements were performed with the usual "2$\tau$ analysis" using 1% temperature rise and 2$\tau$ measurement time. The temperature increment was 0.5 K in the temperature range 2.0–15 K, then 1 K increment between 16 and 100 K, finally 2 K increment above. For the single pulse experiments, a temperature rise of 4 K was targeted, which was found sufficient to fully cover the transition range in zero magnetic field, but only partially covers the transition widths at higher magnetic fields.

**X-ray absorption spectroscopy**. X-ray absorption and magnetic circular dichroism experiments were carried out at the ID12 beamline of the European Synchrotron Radiation Facility, ESRF, France[48]. The experimental end station is equipped with a cold finger cryostat allowing a control of the temperature in the range 2.1–300 K, as well as magnetic field up to 17 T. The x-ray absorption spectra were recorded in fluorescence with a Si photodiode covered with polypropylene film and mounted in backscattering geometry. The x-ray absorption and consequently XMCD spectra were corrected for self-absorption effects. The XMCD spectra were obtained as difference between x-ray absorption spectra with opposite helicities of the incoming x-rays, and for two opposite directions of the magnetic field.

**Computational details**. First-principles electronic structure calculations were performed using the experimental crystal parameters. The local spin density approximation including onsite 4$f$ electron correlation[49] and spin orbit coupling (LSDA + U + SOC) approach has been employed. This approach is implemented in the tight binding linear muffin tin orbital (TB-LMTO, within the atomic sphere approximation, ASA) and full potential linear augmented plane wave (FP-LAPW) band structure methods[50,51]. Both approaches yield identical results, and those reported here are derived from TB-LMTO-ASA. Electronic structure calculations performed with different values of onsite 4$f$-electron correlation parameter ($U$) ranging from 1 eV to 7 eV. The results shown here are with $U$ = 6.7 eV and onsite 4$f$-electron exchange parameter $J$ = 0.7 eV, which are known parameters for Gd-based systems. The basis sets consisted of Eu-$s$, $p$, $d$, $f$ and In-$s$, $p$, $d$ orbitals. The calculated 4$f$ and 5$d$ orbital moments in $Eu_2In$ are within ~0.03 $\mu_B$, and were thus neglected. The k-space integrations have been performed with 16 × 16 × 16 Brillouin zone mesh, which was sufficient for the convergence of total energies and magnetic moments.

**Data availability**. The data that support the findings of this study are available from the corresponding author F.G. upon reasonable request.

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

## Acknowledgements

The Ames Laboratory is operated for the U.S. Department of Energy (DOE) by Iowa State University of Science and Technology under contract No. DE-AC02-07CH11358. This work was supported by the Office of Science of the U.S. DOE, Division of Materials Sciences and Engineering, Office of Basic Energy Sciences. We thank T. Hackett for his

preliminary electronic structure calculations. We acknowledge the European Synchrotron Radiation Facility for provision of synchrotron radiation at the ID12 beamline.

## Author contributions

All authors contributed equally to this work, discussed the results and commented on the manuscript at all stages. The work was initiated by V.K.P and F.G. F.G. carried out the sample synthesis and physical characterizations, A.K.P. contributed to the high pressure magnetization measurements, Y.M. the temperature dependent XRD, D.P. conducted the first principle calculations, F.W. and A.R. the XAS/XMCD measurements. The work was supervised by V.K.P.

## Additional information

**Competing interests:** The authors declare no competing interests.

