## [Peer Review File · Nature Communications]

Reviewers' comments:

Reviewer #1 (Remarks to the Author):

Reviewer report concerning the manuscript „A nearly perfect non-hysteretic first-order phase transition with extremely large latent heat and giant low-field magnetocaloric effect” by F. Guillou et al., your ref. no.: NCOMMS-18-04263-T

The current manuscript reports on new and original investigations on a binary compound, Eu₂In, that belong to the class of materials that show a magneto-elastic first-order magnetic transition. In the case of Eu₂In, the transition is between the paramagnetic – and the ferromagnetic state. Such materials are at the same time rare and desirable due to applications e.g. in magnetic refrigeration. The special case for these materials is that the magnetic transition involves a latent heat (as required by the definition of a first order phase transformation according to Ehrenfest’s classification) while in most cases the transformation is of second order. Additionally, this specific group of materials undergoes the transition without change of the symmetry of the crystal lattice. In the current manuscript including the supplementary information, the authors show in a clear and consistent manner that the transformation in the Eu₂In compound is a phase transformation between paramagnetic and ferromagnetic states that does not involve a change of the symmetry of the crystal lattice. Moreover, in-situ XRD analyses confirmed that the transformation did not involve a large volume change nor does it involve large anisotropic distortions. Moreover, by dedicated calorimetric measurements, the authors could also unambiguously show that the transformation is of first order and involves a remarkably large latent heat and a very small thermal hysteresis.

Analyses of the electronic structure by experiments and theoretical calculations indicate that the underpinning mechanism for the unusual transformation is related to the filling of the Eu-5d states by hybridization with other Eu states and In-5p states, and the development of a large exchange splitting on Eu-5d and In-5p states. The strong hybridization between Eu-5d and In-p states near the Fermi energy affects the long range exchange interaction between Eu atoms, triggering the ferromagnetic transition.

Thus, the present manuscript reports on a new and highly interesting first-order magnetic transition. With the insight gained by the present work, it seems plausible and possible to design new materials that show magneto-elastic first-order magnetic transitions.

The manuscript in its present state is new, original and highly interesting. At the same time, it is ordered in a consistent and logical way. The information given is sufficiently detailed to assess the quality of the experimental and theoretical analyses. The results are of high interest for the entire community that works on magnetic phase transformations and is of special importance for those who are interested in magneto-caloric materials.

With this stated, I recommend accepting the current manuscript in its present state for publication.

Reviewer #2 (Remarks to the Author):

The manuscript by Guillou et al. presents experimental evidence of a first-order phase transition with large latent heat and magnetocaloric effect in Eu₂In. The transition occurs around 55 K and shows exceptional characteristics in terms of magnetocaloric parameters (large entropy change, small applied field, small hysteresis, etc.). The authors provide a simple electronic structure explanation of the observed behavior. The manuscript is nicely prepared and puts forward results which are of interest for people working within the field. On the other hand, the practical importance of the disclosed phenomenon is not clear. Although I find the present research and

results of very high quality and interesting primarily for specialists, they might not be of direct interest to the broad audience of Nature Communication.

Some specific comments:

- 1) The abstract and introductory part discusses very lengthily the nature of magnetic transitions, turning to the actual subject only after approx. 2 pages of introduction. The authors might wish to reorganize and shorten the general introduction in order to make the paper more focused. The need for Fig. 1 is not clear either.
- 2) In the main text, the measured lattice parameters correspond to room-temperature (I guess) which should be stated.
- 3) The validity of the anti-ferromagnetic type modeling of the paramagnetic state is unclear. This should be either motivated by other approximations/alternative models or removed from the main text.
- 4) They adopt LDA+U+SOC ab initio method which is implemented in TB-LMTO and FP-LAPW (as stated in the manuscript). But they do not reveal whether they used LMTO (perhaps ASA) or LAPW (FP).
- 5) It is unclear why the basis set was truncated at s,p orbitals for In. It is not impossible but seems to be a totally unnecessary approximation.
- 6) Calculations were performed as a function of the Coulomb/exchange parameter U/J, but the results are presented only for 6.7 eV/0.7 eV. This further questions the validity of the theoretical modeling. Can one call the approach really first-principles? How would the transition be affected by the particular U/J values, how the optimal U/J depend on the magnetic state.

Reviewer #3 (Remarks to the Author):

F. Guillou, A.K. Pathak, D. Paudyal, Y. Mudryk, F. Wilhelm, A. Rogalev, and V.K. Pecharsky present extensive experimental and electronic structural evidence and manifestation of "A nearly perfect non-hysteretic first-order phase transition with extremely large latent heat and giant low-field magnetocaloric effect." This work is both of scientific and technical interest in that it demonstrates:

1. A material in which the first order magnetocaloric phase transition (FOMT) has origins in a rare isosymmetric FOMT. This is borne out by careful experimentation and in and of itself is a very interesting scientific finding.
2. The isosymmetric FOMT is significant in that it gives rise to a low field and non-hysteretic giant magnetocaloric effect. Such transitions are important for technically useful magnetocaloric materials.

However, the transition occurs at 55 K and most research on technically important magnetocaloric materials target room temperature applications. Nevertheless, the work is so interesting, well performed and important that publication in Nature is warranted. I would ask the authors to consider some additional discussion of the following:

1. $T_c(B)$ (Fig. 2b indicates measured values of 62 K and extrapolated values above 77 K. Can the authors speculate on cryogenic applications of the magnetocaloric effects near LN2 boiling temperatures.
2. While large peak entropy changes are typical figures of merit for magnetocaloric materials, useful magnetocaloric materials are further discussed in terms of refrigeration capacity (coefficient). In this respect, the data of Figure 2c is of interest. Some discussion of the broadening under pressure is warranted.
3. If the MCE is mediated by d-electron exchange interactions its magnitude and distribution will influence such broadening and may be discussed in terms of distributed exchange interactions and the Bethe-Slater curve. See:
M. Kurniawan, A. Perrin, P. Xu, V. Keylin, and M. E. McHenry, "Curie Temperature Engineering in

High Entropy Magnetic Alloys for Magnetocaloric Applications," IEEE Mag. Lett. 7, 1-5, (2016). doi: 10.1109/LMAG.2016.2592462.

N. J. Jones, H. Ucar, J. J. Ipus, M.E. McHenry, and D. E. Laughlin, "The Effect of Distributed Exchange Parameters on Magnetocaloric Refrigeration Capacity in Amorphous and Nanocomposite Materials." J. Appl. Phys. 111, 07A334-336 (2012). (<http://dx.doi.org/10.1063/1.3679456>).

The former addresses broadening due to positional disorder on a crystalline lattice, the latter positional disorder in amorphous materials in the context of broadening of a 2nd order magnetic transition. Are similar ideas appropriate for an FOMT?

4. The low T_c of these materials seems to preclude typical commercial refrigeration applications but applications in space seem relevant. Please comment.

Reviewer #1

No critique to address, and we very much appreciate a positive assessment.

Reviewer #2

We thank the reviewer for a positive assessment and note that the mechanism leading to an isosymmetric first-order magnetic transition we describe is novel. Not only this carries fundamental importance, but excellent magnetocaloric-relevant parameters associated with both the transition and the mechanism, open doors to future applications. Though limited, for the moment, to the vicinity of the ferromagnetic transition in Eu_2In at 55 K, we are convinced our report will motivate further studies, feasibly extending the domain of applications toward room temperature.

Specific Comments:

1) *The abstract and introductory part discusses very lengthy the nature of magnetic transitions, turning to the actual subject only after approx. 2 pages of introduction. The authors might wish to reorganize and shorten the general introduction in order to make the paper more focused. The need for Fig. 1 is not clear either.*

The introduction, pp. 2 and 3, has been shortened by 160 words (~30%), as marked. We believe Figure 1 appeals to a broad audience by presenting a layman classification of magnetic phase transitions with a clear representation of the branch relevant for caloric applications. Hence, we would like to keep it.

2) *In the main text, the measured lattice parameters correspond to room-temperature (I guess) which should be stated.*

Done on p. 5.

3) *The validity of the anti-ferromagnetic type modeling of the paramagnetic state is unclear. This should be either motivated by other approximations/alternative models or removed from the main text.*

Though more advanced methods exist, non-polarized DFT and/or imposed zero *net* moment antiferromagnetic solution are two common approaches frequently reported in the literature to model paramagnetic states. Since Eu *4f* moments are localized, non-spin polarized calculations would not be relevant, nor realistic. Presence of two inequivalent Eu sites with identical multiplicities in Eu_2In makes an easy approximation of the paramagnetic state by forcing two independent Eu sublattices antiparallel. The result is a negligible ($\sim 0.05 \mu_B$ /unit cell) *net* magnetic moment. Strictly speaking, this is not a true magnetically disordered state, but the advantage of such approach is in preservation of the *local* *4f* moments of Eu. As requested, AFM DOS and relevant discussion have been moved into the supplementary information,

item J; this change mandated a minor revision as marked toward the top of p. 13.

4) They adopt LDA+U+SOC *ab initio* method which is implemented in TB-LMTO and FP LAPW (as stated in the manuscript). But they do not reveal whether they used LMTO (perhaps ASA) or LAPW (FP).

LDA+U+SOC calculations were performed using TB-LMTO within atomic sphere approximation (ASA). In order to check the validity of ASA for Eu_2In , FP-LAPW calculations were also performed. The results were identical. Both details are added into the revised manuscript in the methods section, p. 18.

5) It is unclear why the basis set was truncated at *s,p* orbitals for In. It is not impossible but seems to be a totally unnecessary approximation.

Indeed, the occupied 4*d* orbitals of In have been included in the basis set. The typo has been corrected.

6) Calculations were performed as a function of the Coulomb/exchange parameter U/J , but the results are presented only for 6.7 eV/0.7 eV. This further questions the validity of the theoretical modeling. Can one call the approach really first-principles? How would the transition be affected by the particular U/J values, how the optimal U/J depend on the magnetic state.

The outcome of our electronic structure calculations, in particular about the treatment of Eu-4*f* states, is supported by x-ray photoelectron spectroscopy (a new panel in Figure 5 and relevant discussion are added on pp. 13 and 14 together with a new reference 48). As also shown below, the values of U in the range of 4 to 7 eV result in approximately constant magnetic moments, further supporting the notion that U/J known for Gd (Ref. 53) are applicable to Eu_2In . Actual value of U has no effect on the ground state, which remains ferromagnetic for all U .

Figure: Variation of magnetic moments as a function of U , for Eu1 and Eu2 in the FM and AFM states, and for In moments in the FM state (In moments are negligible in the AFM state). X (-1) denotes the antiparallel Eu2 moments.

Reviewer #3

1. $T_c(B)$ (Fig. 2b indicates measured values of 62 K and extrapolated values above 77 K. Can the authors speculate on cryogenic applications of the magnetocaloric effects near LN2 boiling temperatures.

The caption of Figure 2 has been modified to make clear that the shift of the transition to ~80 K in high magnetic field is not an extrapolation but an experimental observation derived from the measured $M(T)_{B = \text{const}}$ curves. Magnetic refrigeration indeed has the potential for gas liquefaction, and this is now mentioned in the introduction, p. 2.

2. While large peak entropy changes are typical figures of merit for magnetocaloric materials, useful magnetocaloric materials are further discussed in terms of refrigeration capacity (coefficient). In this respect, the data of Figure 2c is of interest. Some discussion of the broadening under pressure is warranted.

Two of the most recently proposed figures of merit are included in the revised manuscript on p. 9 (this change also required adding two new references, Refs. 32 and 33). A sentence briefly discussing the broadening has been added on p. 6.

3. If the MCE is mediated by d-electron exchange interactions its magnitude and distribution will influence such broadening and may be discussed in terms of distributed exchange interactions and the Bethe-Slater curve. See: M. Kurniawan, A. Perrin, P. Xu, V. Keylin, and M. E. McHenry, "Curie Temperature Engineering in High Entropy Magnetic Alloys for Magnetocaloric Applications," *IEEE Mag. Lett.* 7, 1-5, (2016). doi: 10.1109/LMAG.2016.2592462.

N. J. Jones, H. Ucar, J. J. Ipus, M.E. McHenry, and D. E. Laughlin, "The Effect of Distributed Exchange Parameters on Magnetocaloric Refrigeration Capacity in Amorphous and Nanocomposite Materials." *J. Appl. Phys.* 111, 07A334-336 (2012). The former addresses broadening due to positional disorder on a crystalline lattice, the latter positional disorder in amorphous materials in the context of broadening of a 2nd order magnetic transition. Are similar ideas appropriate for an FOMT?

A sentence addressing this comment (p. 7), two references indicated by the reviewer (Refs.28 and 29), as well as our understanding of other potential influences that can be linked to the extreme sharpness of the observed FOMT have been added.

4. The low T_c of these materials seems to preclude typical commercial refrigeration applications but applications in space seem relevant. Please comment.

Done, also see our response to the first comment.

REVIEWERS' COMMENTS:

Reviewer #2 (Remarks to the Author):

The authors have addressed the questions and observations from the reports. The changes are appreciated. Although I miss the practical impact of this work and also the broad interest in the disclosed phenomenon, I consider the work scientifically of high quality and support publication.